# Effect of Hole Shift on Threshold Characteristics of GaSb-Based Double-Hole Photonic-Crystal Surface-Emitting Lasers

**DOI:** 10.3390/mi12050468

**Published:** 2021-04-21

**Authors:** Yu-Hsun Huang, Zi-Xian Yang, Su-Ling Cheng, Chien-Hung Lin, Gray Lin, Kien-Wen Sun, Chien-Ping Lee

**Affiliations:** 1Institute of Electronics, National Yang Ming Chiao Tung University, Hsinchu City 30010, Taiwan; a123452001@gmail.com (Y.-H.H.); kkobeandmek@gmail.com (Z.-X.Y.); suling0115@gmail.com (S.-L.C.); cplee@mail.nctu.edu.tw (C.-P.L.); 2R&D Division, Phosertek Corporation, Hsinchu City 30010, Taiwan; tony.lin@phosertek.com; 3Department of Applied Chemistry, National Yang Ming Chiao Tung University, Hsinchu City 30010, Taiwan; kwsun@mail.nctu.edu.tw

**Keywords:** GaSb-based lasers, photonic crystals, surface-emitting lasers, infrared lasers

## Abstract

Photonic-crystal (PC) surface-emitting lasers (SELs) with double-hole structure in the square-lattice unit cell were fabricated on GaSb-based type-I InGaAsSb/AlGaAsSb heterostructures. The relative shift of two holes was varied within one half of the lattice period. We measured the lasing wavelengths and threshold pumping densities of 16 PC-SELs and investigated their dependence on the double-hole shift. The experimental results were compared to the simulated wavelengths and threshold gains of four band-edge modes. The measured lasing wavelength did not exhibit switching of band-edge mode; however, the calculated lowest threshold mode switched as the double-hole shift exceeded one quarter of the lattice period. The identification of band-edge lasing mode revealed that modal gain discrimination was dominated over by its mode wavelength separation.

## 1. Introduction

Photonic-crystal (PC) surface-emitting lasers (SELs), incorporating two-dimensional (2D) and second-order grating structures over broad area, exhibit the advantages of high output power, narrow spectral linewidth and small beam divergence [1,2,3]. Promising applications include material processing, spectroscopic sensing, light detection and ranging. Recently, engineering of 2D-PC has evolved from single-hole structure towards double-hole structure in order to achieve high brightness emissions and maintain spatially single-mode operation within the emission diameter of a millimeter scale [2].

GaSb-based semiconductors are ideal candidates for mid-infrared (MIR) light emitters of 2–5 μm [4]. However, the lasing wavelengths of high performance lasers composed of type-I InGaAsSb/AlGaAsSb quantum-well (QW) heterostructures are not beyond 3 μm or, in particular, below 2.5 μm [5]. In recent years, epitaxial laser structures on GaSb substrates became the platform for our PC-SEL study [6,7,8,9]. There are at least three advantages that can be addressed. First, the longer emission wavelength means longer lattice period and larger PC holes, which relax the process conditions of e-beam lithography and dry etching. Second, pulsed fiber laser of 1064 nm is readily available for optically pumped lasing emissions longer than 1 μm. Third, top-cladding layer thickness is about an order-of-magnitude thinner for PC-SELs as compared to edge emitting lasers.

Previously, we have systematically investigated the effect of etching depth on the threshold characteristics of GaSb-based PC-SELs with single-hole PC structure [9]. In this work, we study the effect of hole shift on the threshold characteristics of double-hole PC-SELs based on GaSb substrate. The measured lasing wavelengths are compared to the simulated wavelengths of four band-edge modes. Besides, the experimental threshold pumping power densities are also examined by the simulated lowest threshold gains among band-edge modes. Not only the threshold gain but also its wavelength dependence should be considered in the identification of lasing band-edge mode.

## 2. Materials and Methods

The investigated sample was grown on n-type GaSb (001) substrates using a Veeco GEN II molecular beam epitaxy system. Figure 1a shows the schematic structure. It consisted of, from the bottom, a 200-nm GaSb buffer layer, a 2000-nm Al_0.85_Ga_0.15_As_0.07_Sb_0.93_ bottom-cladding layer, a 150-nm Al_0.3_Ga_0.7_As_0.02_Sb_0.98_ separate confinement layer (SCL), four layers of 12-nm In_0.35_Ga_0.65_As_0.14_Sb_0.86_ quantum well (QW) spaced by three layers of 25-nm Al_0.3_Ga_0.7_As_0.02_Sb_0.98_ layer, a 200-nm Al_0.3_Ga_0.7_As_0.02_Sb_0.98_ SCL, a 200-nm top-cladding layer Al_0.5_Ga_0.5_As_0.04_Sb_0.96_ and a 20-nm GaSb capping layer. The room-temperature (RT) photo-luminescence (PL), shown in Figure 1b, peaked around 2330 nm with a full width at half maximum (FWHM) of about 100 nm. The region of lasing emissions was designed in the high-energy side of the PL and located just outside its spectral FWHM.

The PC used in our devices had circular air holes arranged in a square lattice of 290 × 290 µm^2^. Two types of PC structures were studied in this work. One had a single-hole structure for the unit cell (Figure 2a). The other had a double-hole structure (Figure 2b) with the two circular holes separated in *x* and *y* directions by the same distance of Δ. The lattice constant or period “*a*” was the same for both cases. The radii of the holes for the single-hole case are *r*_1_, and *r*_2_ for the double-hole case. The fabrication process details can be found in [8].

The lattice period was fixed at 650 nm and the designed *FF* (filling factor, fraction of the hole area in a unit cell) was 10%. The hole radii of *r*_1_ and *r*_2_ were therefore calculated to be 116 nm and 82 nm, respectively. Table 1 lists all the parameters of the fabricated devices. The 16 devices were processed at the same time and in a small piece of wafer. Device A was the single-hole PC-SEL, and Devices B to P were double-hole PC-SELs with 15 different hole spacings. Notice that the two holes in the unit cell overlap if the shift distance is smaller than 120 nm and the *FF* is lower than 10%. In this design, the single-hole structure is a special case of the double-hole structure with zero shift (Δ = 0 nm).

The plan-view scanning electron microscope (SEM) images of 16 PC-SELs are shown in the Appendix A. The hole radii and *FF*s were measured by the Image-Pro (Media Cybernetics, Inc., Rockville, MD, USA) as shown in Table 1. Figure 3 shows the cross-sectional SEM images of the single-hole structure (Device A) and the double-hole structure with a shift distance of 160 nm (Device I). The etched depths were 392 nm and 311 nm for single-hole and double-hole structures, respectively. The difference was due to the loading effect during dry etching.

## 3. Results and Discussions

The devices were temperature controlled and optically pumped by a 1064-nm pulsed fiber laser. The experimental setup and measurement conditions for surface emitting lasers were detailed in [9]. We use Device A and Device I as examples to show their lasing characteristics. Figure 4 shows the lasing spectra and light-in versus light-out (*L-L*) characteristics. The lasing wavelengths were 2252.5 nm and 2235.2 nm for Devices A and I, respectively. Their spectral FWHM were both less than 0.5 nm. The threshold pumping power densities for Device I (1.91 KW/cm^2^) were more than five times higher than those of Device A (0.36 KW/cm^2^). Nevertheless, Device I shows an enhanced slope efficiency, which is more than two times higher than that of Device A. Since output intensities are sensitive to pumping and collection conditions, slope efficiencies may have large deviation among measurements; therefore, we have no intension to compare the relative slopes of the devices.

The emission wavelengths of 16 devices are plotted against the relative shift ratio and shown in Figure 5a. The dashed fitting curve is cosine-like and the shortest lasing wavelength was achieved for Device I (with a shift ratio about 1/4). To uncover the dependence of lasing wavelength on the shift ratio, we have carried out the two-dimensional (2D) photonic band-structure calculation by plane wave expansion (PWE) using the software from RSoft Design Group. The simulated band-structures are shown in the Appendix A. There are four band-edge modes near the Γ point and conventionally designated, in order of decreasing wavelength, as ModeA, ModeB, ModeC and ModeD. The four band-edge wavelengths are plotted against the shift ratio as shown in Figure 5b. We can therefore infer that ModeA or ModeB is the most probable lasing mode by comparing Figure 5a,b.

To further investigate the dependence of pumping threshold on shift ratio, we have implemented the three-dimensional (3D) coupled-wave theory (CWT) model for square-lattice PC-SELs based on Noda’s papers [10,11]. Simulated devices are finite in size (300 × 300 μm^2^) and double-hole structures without overlap (Δ/*a* ≥ 0.18) are considered only. The calculated modal power loss (including in-plane and radiation loss) are taken as the threshold gain. The four band-edge wavelengths and corresponding threshold gains are plotted against shift ratio as shown in Figure 6a,b, respectively. The band-edge wavelengths versus shift ratio in Figure 6a follow the same trend as those in Figure 5b. However, a closer look at Figure 6b reveals that the lowest threshold mode switches from ModeA to ModeD as shift ratio is exceeding 1/4. If the lowest threshold mode is the lasing mode, the lasing wavelength should be monotonically decreasing with respect to the shift ratio, which is inconsistent with Figure 5a.

The threshold pumping power densities of 16 devices are plotted against respective shift ratio and shown in Figure 7. We have superimposed in Figure 6 the simulated lowest threshold gains of ModeA or ModeD. The measured threshold is quantitatively well fitted by the simulated threshold of ModeA. The arguments that ModeA rather than ModeD is favored at a large ratio exceeding 1/4 are twofold. First, the lasing wavelengths are in the short-wavelength side of the PL spectrum and located outside its FWHM. The longer wavelength of ModeA is therefore favored because the ModeD wavelength is shorter for several tens of nanometers. Second, the in-plane interference is totally destructive around the shift ratio of 1/4. The required pumping threshold is greatly enhanced and ModeA is still favored even the wavelength separation between ModeA and ModeD is a little less than 10 nm. Therefore, wavelength-dependent gain characteristics dominated over modal gain discrimination and ModeA lasing emission was achieved. For the reader’s interest, the simulated radiation field as well as far-field intensity distributions of ModeA and ModeD for three different shift ratios (0, 0.25 and 0.5) are shown in the Appendix A.

## 4. Conclusions

We have fabricated GaSb-based type-I InGaAsSb/AlGaAsSb PC-SELs with double-hole design in the square-lattice unit cell. The threshold characteristics of emission wavelength and pumping density were systematically investigated on the shift distance of double holes. The lasing wavelength was oscillated within the half-period shift of double holes. The shortest wavelength was achieved around the quarter-period shift of double holes where in-plane interference was totally destructive. The PWE and CWT simulations were performed and compared to our experimental results. As the double-hole shift exceeded one quarter of the lattice period, the calculated lowest threshold mode switched from ModeA to ModeD. However, the measured threshold pumping density was analyzed to lase from ModeA and consistent with the analysis from the measured wavelength. It can be argued that the wavelength-dependent gain characteristics dominated and rendered the lasing emission to be ModeA.

## Figures and Tables

**Figure 1 micromachines-12-00468-f001:**
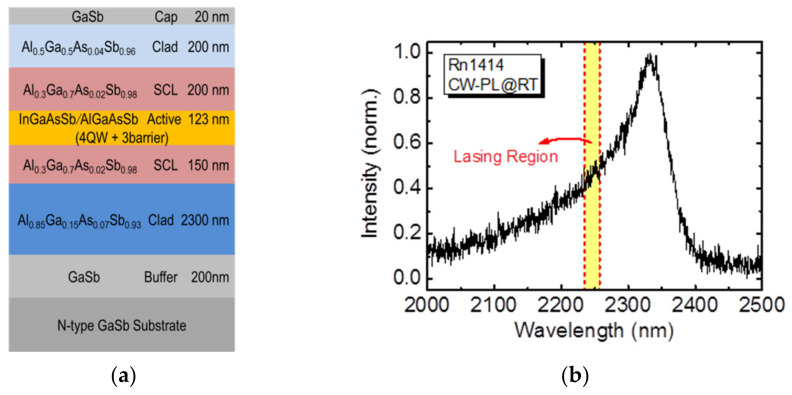
(**a**) The schematic of the epitaxial layer structure and its (**b**) room-temperature photo-luminescence.

**Figure 2 micromachines-12-00468-f002:**
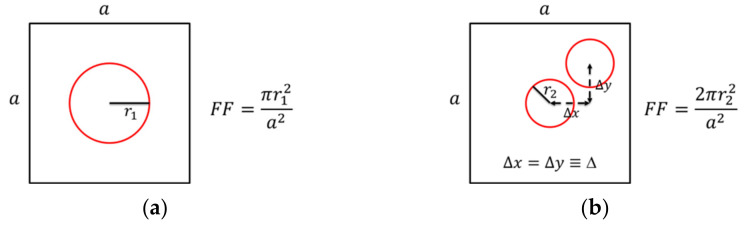
The unit cells of (**a**) single-hole and (**b**) double-hole PC structures.

**Figure 3 micromachines-12-00468-f003:**
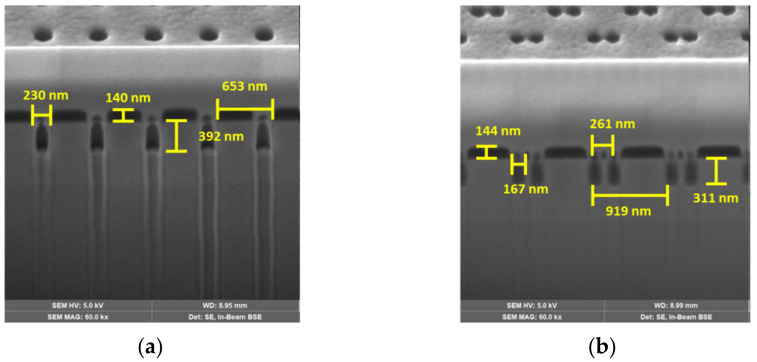
The cross-sectional SEM images of (**a**) single-hole PC structure (Device A) and (**b**) double-hole PC structure with shift distance of 160 nm (Device I).

**Figure 4 micromachines-12-00468-f004:**
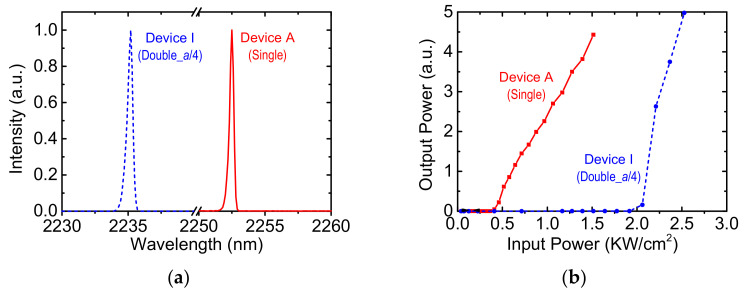
The optically pumped (**a**) lasing spectra and (**b**) *L-L* characteristics for Device A and I.

**Figure 5 micromachines-12-00468-f005:**
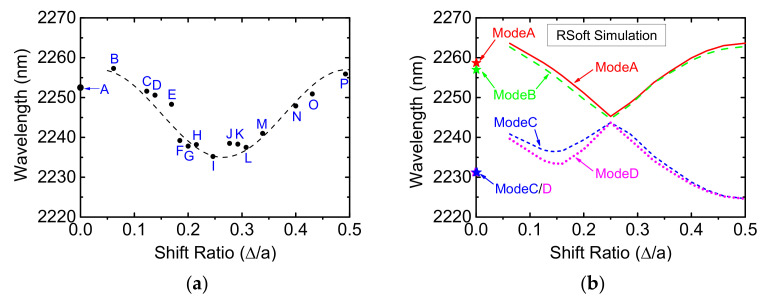
(**a**) The lasing wavelength versus shift ratio for 16 experimental devices. (**b**) The four band-edge wavelengths versus shift ratio by RSoft simulation.

**Figure 6 micromachines-12-00468-f006:**
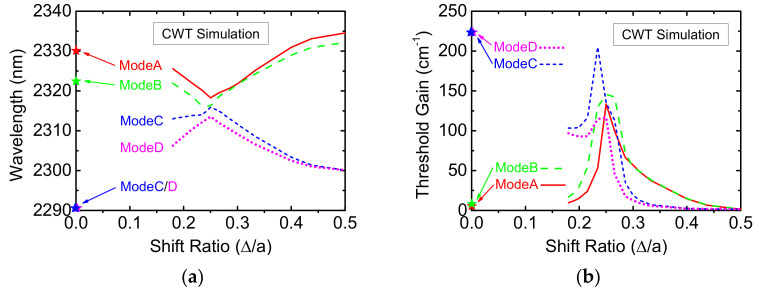
The simulated (**a**) band-edge wavelengths and corresponding (**b**) threshold gains are plotted versus shift ratio based on the CWT model.

**Figure 7 micromachines-12-00468-f007:**
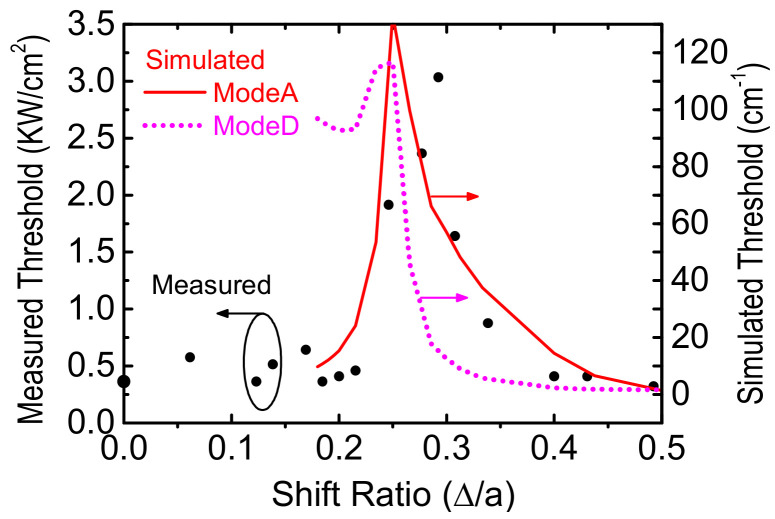
The threshold power density versus shift ratio for 16 experimental devices. The simulated threshold gains of ModeA and ModeD are superimposed.

**Table 1 micromachines-12-00468-t001:** The single-hole PC-SEL as well as double-hole PC-SELs with 15 shift distances.

Device No.	Designed Δ (nm)	Measured *r*_1_/*r*_2_ (nm)	Measured *FF* (%)
A	0	104.8	8.4
B	40	81.8	6.8
C	80	80.3	8.8
D	90	79.6	8.3
E	110	81.9	9.2
F	120	81.8	9.7
G	130	81.0	9.7
H	140	83.3	10.9
I	160	79.9	9.7
J	180	77.9	9.4
K	190	77.6	9.5
L	200	80.9	10.2
M	220	80.6	10.0
N	260	76.9	9.1
O	280	78.5	9.6
P	320	83.0	10.5

## Data Availability

The data that support the findings of this study are available from the corresponding author upon reasonable request.

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
