# Peer review of "Effect of Hole Shift on Threshold Characteristics of GaSb-Based Double-Hole Photonic-Crystal Surface-Emitting Lasers"

_micromachines, 2021, doi:10.3390/mi12050468_

Round 1

Reviewer 1 Report

The authors present work concerning photonics crystal surface emitting lasers, with emission in infrared range of spectrum, ~2.2 microns. The work is written clearly, the results are presented with sufficient theoretical explanations. 
The appendices are useful and helpful in understanding of the problem. Moreover, SEM images present very high quality of fabrication methods.
The strong side of the work is that design part is backed-up by experiment: namely the devices are fabricated and characterised. The choice of wavelength indeed decreases the technological considerations for fabrication, but on the other hand this spectral region is interesting from the point of view of applications, especially in gas detection.
The subject is potentially interesting to community of laser specialists and theoreticians, especially, that work covers increasingly important spectral range.
I recommend the paper for publication.

Author Response

Thanks for your encouraging comments.

Reviewer 2 Report

The authors study (experimentally and through simulations) he effect of double hole separation on the lasing threshold and wavelength of GaSb based PC-SEL. The study is clear and consistent. Few minor recommendations are provided.

Page 7 line 131: Change Figure 6 to Figure 7.

Minor: Plot the field distribution of mode A and D at three different shift ratios (say 0, 0.25 and 0.5, or similar)

Minor: Make a comment on the lasing polarization direction and its dependence on the unit cell orientation.

Author Response

Thanks for your suggestions. We have corrected the mistake. For more information on field distribution and polarization, we add the third section “Radiation field distributions, far-field patterns and polarizations” in the Supplementary Materials. The following texts are also mentioned in the revised manuscript.

(in paragraph above Figure 7)

For your interest, the simulated radiation field as well as far-field intensity distributions of ModeA and ModeD for three different shift ratios (0, 0.25 and 0.5) are shown in the Supplementary Materials (Figure S3).